# Associations between Fatty Acid Intake and Tension-Type Headache: A Cross-Sectional Study

**DOI:** 10.3390/jcm11237139

**Published:** 2022-12-01

**Authors:** Diego Domínguez-Balmaseda, José Ángel Del-Blanco-Muñiz, Angel González-de-la-Flor, Guillermo García-Pérez-de-Sevilla

**Affiliations:** 1Department of Physiotherapy, Faculty of Sport Sciences, Universidad Europea de Madrid, 28670 Villaviciosa de Odón, Spain; 2Masmicrobiota Group, Faculty of Health Sciences, Universidad Europea de Madrid, 28670 Villaviciosa de Odón, Spain

**Keywords:** tension-type headache, monounsaturated fatty acids (MUFAs), polyunsaturated fatty acids (PUFAs), Omega-3 (ω-3), Omega-6 (ω-6)

## Abstract

Introduction: Patients with tension-type headache (TTH) are characterized by recurrent pain that can become disabling. Identifying the dietary triggers of headaches has led to defining dietary strategies to prevent this disease. In fact, excessive dietary intake of Omega-6 (ω-6) fatty acids, or an ω-6: ω3 ≥ 5 ratio, typical of Western diets, has been associated with a higher prevalence of headaches. The objectives of the present study were to compare dietary fatty acid intake between participants with and without chronic TTH and to investigate the association between dietary fatty acid intake, pain characteristics, and quality of life in patients with chronic TTH. Methods: An observational study was conducted, comparing healthy participants (*n* = 24) and participants diagnosed with chronic TTH for more than six months (*n* = 24). The variables analyzed were dietary fatty acid intake variables, the Headache Impact Test (HIT-6), and the characteristics of the headache episodes (intensity, frequency, and duration). Results: The TTH group reported a significantly higher intake of saturated fatty acids (SFAs) but similar intakes of monounsaturated fatty acids (MUFAs), polyunsaturated fatty acids (PUFAs), and ω-6: ω-3 ratio when compared to controls. Furthermore, in the TTH group, the Ω-6 fatty acid intake was associated with more intense headache episodes. In addition, the TTH group reported a significant impact of headaches on their activities of daily living according to the HIT-6. Conclusions: Higher intakes of SFAs and Ω-6 fatty acids were associated with more severe headache episodes in patients with TTH. Therefore, the characteristics of the diet, in particular the dietary fatty acid intake, should be considered when treating these patients.

## 1. Introduction

Tension-type headache (TTH) is characterized by recurrent headaches that can become disabling, often accompanied by sensory, motor, and neurovascular alterations, with a prevalence of 26% in general population [1,2].

Although the cause of TTH is unclear and multifactorial, the pathophysiology of migraine involves immune response factors and oxidative stress [3,4], which could lead to neuronal inflammation [2,5]. The pathogenesis and progression of TTH appear to be influenced by a combination of physiological, psychosocial, and lifestyle factors [6,7]. Some of these factors, including nutritional patterns, are amenable to behavioral and clinical interventions [8]. Recently, TTH has been associated with dietary elements [9]. In fact, advances in understanding the mechanisms underlying headache pathogenesis and how dietary factors may interfere with those mechanisms have encouraged investigators to consider diet as a disease-modifying agent [10].

Identifying the dietary triggers of headaches has led to defining dietary strategies to prevent this disease [11]. For example, a higher intake of alcohol or canned foods rich in nitrates and nitrites, monosodium glutamate, and artificial sweeteners has been linked to a higher prevalence of headaches [12]. In addition, excess fat intake is widely accepted as unhealthy, but the type of fat intake is more clinically relevant [13].

Fatty acids can be classified depending on their length and degree of saturation into saturated fatty acids (SFAs), monounsaturated fatty acids (MUFAs), and polyunsaturated fatty acids (PUFAs). Among essential PUFAs, Omega-3 (ω-3) fatty acids, such as eicosatetraenoic acid (EPA) and docosahexaenoic acid (DHA), are crucial for the nerve system’s functioning. Several studies have shown the effects of anti-inflammatory and neuroprotective effects of the ω-3 fatty acids [14,15]. On the contrary, excess of Omega-6 (ω-6) fatty acids dietary intake, or an ω-6: ω3 ratio ≥ 5, which is typical in western diets, has been associated with a higher prevalence of headaches [10,16]. In addition, a higher dietary intake of ω-3 can reduce the frequency of headaches, according to a recent study [17]. Similarly, excess SFAs intake could favor inflammatory reactions in some tissues and could be responsible for causing headaches, on the contrary, an adequate intake of polyunsaturated fatty acids could lead to an improvement in the quality of life of patients with TTH, preventing the headache or reducing it considerably [9], as well as the intensity [17] (Figure 1).

Therefore, diet must be considered a relevant modifiable factor that needs more attention. Given the paucity of research about this topic and gaps in knowledge, the aims of the present study were: (i) To compare the dietary intake of fatty acids between participants with and without chronic TTH, (ii) to investigate the cross-sectional association between fatty acids dietary intake, pain characteristics, and quality of life in patients with chronic TTH patients.

## 2. Methods

### 2.1. Study Design

An observational study following the Strengthening the Reporting of Observational Studies in Epidemiology (STROBE) Initiative Statement [18], comparing healthy participants and participants with diagnosed chronic TTH for more than six months according to the criteria of the International Headache Society (IHS) [19].

The study protocol adhered to the principles of the 1964 Declaration of Helsinki and its subsequent clarifications and was approved by the Research Ethics Committee of the Rey Juan Carlos University of Madrid (reference number: 1802202105721).

### 2.2. Participants

Recruitment of participants was carried out (between September and December 2021) among students and workers at the European University of Madrid. Forty-eight participants were recruited for the study, 24 were selected for the control group and 24 were selected to be part of the chronic TTH group. This last group previously attended their medical center, where they were diagnosed with TTH by their neurologist, following the headache classification criteria of the HIS classification in its third edition [19]. The diagnostic criteria followed were: (i) Headache occurs on average more than 15 days per month for more than 3 months (more than 180 days per year) with pericranial tenderness in the neck and face region, (ii) with a duration of minutes to days or without remission, (iii) bilateral mild to moderate pain intensity, and (iv) no more than one of photophobia, phonophobia, or mild nausea. Inclusion criteria for the TTH group were: (1) Adults aged 18–65 years, (2) TTH for more than six months (diagnosed with chronic TTH by their neurologist, following the criteria of the IHS) [19]. The inclusion criteria for the control group were: No headache. The exclusion criteria for both groups were: Neurological, metabolic diseases, such as diabetes mellitus, cardiovascular diseases, such as arterial hypertension, or severe respiratory diseases, whiplash injury, neck fracture, and mental health disease (anxiety or depression) assessed with Hospital Anxiety and Depression Scale (>8 points in each subscale) [20]. Individuals with TTH were matched with their healthy controls by gender, age, body mass index (BMI), and nutritional energy intake (kcal).

### 2.3. Variables

Anthropometric variables were gender (male and female), height in centimeters (cm), and weight in kilograms (kg). Height was measured with a measuring rod (Ano Sayol SL, Barcelona, Spain) and weight with a mechanical scale (Asimed T2, Barcelona, Spain). BMI was calculated as Weight (kg)/height (m^2^) following Shephard’s protocol [21].

### 2.4. Characteristics of Headache Episodes

The duration of the headaches expressed in hours, the intensity of the tension headache on the numeric pain rating scale from 0 to 10 and the frequency in days of the episodes were measured in each subject [19].

### 2.5. Headache Impact Test

The Headache Impact Test (HIT-6) measures the impact that headaches have on daily activity tasks. Regarding the severity of the impact, ≥60 means very severe impact, 56–59 significant impact, 50–55 moderate impact, ≤49 little impact [22].

### 2.6. Food Frequency Questionnaire

The diet of the participants was analyzed through the use of validated Food Frequency Questionnaires (FFQ) [23]. Data were analyzed using Dietsource 3.0 software (Novartis, Barcelona, Spain) to obtain dietary intake of SFAs (g), MUFAs (g), PUFAs (g), ω-6 fatty acids (g), and ω-3 fatty acids (g), as well as the total energy intake (kcal). Then, the ω-6: ω-3 ratio was calculated using the total intake values of ω-6 and ω-3 [24]. A total of 134-line items assess the dietary intake over the last three months, categorized into 9 major food groups: Starchy food, fruits, cooked and raw vegetables, meat-poultry-fish-eggs, prepared dishes, dairy products, fats, drinks (alcoholic and non-alcoholic), and miscellaneous. The participants reported the frequency of consumption of each food group on the basis of 6 levels of frequency: Rarely or never, one to three times a month, one to two times a week, three to five times a week, one time a day, 2 times or more a day. The frequency of consumption of food items was multiplied by the portion size to calculate the grams of food consumed per day. FFQ have shown good reliability and validity values [25].

### 2.7. Sample Size Calculation

Previously, a pilot study (*n* = 10) had been performed by using the G*Power 3.1.9.2 Software (G*Power ©, University of Dusseldorf, Düsseldorf, Germany) [26]. To calculate the sample size, the mean differences found between groups (*n* = 5, participants with TTH; *n* = 5, control group) in the dietary intake of SFAs (g) (TTH group, 31.3 ± 10.8 and control group, 22.4 ± 10.4), selected as the main variable. In addition, a two-tailed hypothesis with an effect size of 0.84, an alpha error probability of 0.05 with a statistical power of 0.8, and an allocation ratio (N2/N1) of 1 were employed for sample size calculation. Therefore, a total sample size of 48 participants, divided into two groups of 24 participants with TTH and healthy-matched controls without TTH were recruited.

### 2.8. Statistical Analysis

A descriptive analysis was developed for all the participants using mean ± standard deviation (SD). Then, the Shapiro–Wilk test was employed to assess the normality of the variables [27]. For non-parametric variables, the Mann–Whitney U test was conducted, while the independent samples *t*-test was employed to identify significant differences between groups. In addition, to analyze the relationship between continuous variables, the Spearman correlation test and Pearson correlation test were performed for the non-parametric and the parametric variables, respectively. The magnitudes of correlation between continuous variables were qualitatively interpreted using the following criteria: Trivial (r ≤ 0.1), small (r = 0.1–0.3), moderate (r = 0.3–0.5), large (r = 0.5–0.7), very large (r = 0.7–0.9), and almost perfect (r ≥ 0.9) [28].

The statistical significance was set at an alpha level of <0.05. All analyses were conducted using IBM SPSS for Windows (version 25, IBM Corporation, Armonk, NY, USA).

## 3. Results

### 3.1. Characteristics of the Sample

The age, BMI, and nutritional energy intake were analyzed to see if there were no differences between groups in these confounder variables. There were no significant differences between the TTH group (*n* = 24; 20 females) and the control group (*n* = 24; 19 females) regarding the age of the participants (37.58 ± 13.36 versus 42.57 ± 8.11 years; *p* = 0.12), BMI (25.18 ± 4.62 versus 24.99 ± 3.95 kg/m^2^; *p* = 0.88), and nutritional energy intake (2175.08 ± 519.58 versus 1960.48 ± 403.16 kcal; *p* = 0.11). As both groups were homogeneous in these variables, they would not be responsible for possible differences between groups regarding fat intake (Table 1).

### 3.2. Headache Episodes

The characteristics of the headache episodes and their impact on daily living tasks were assessed only in the TTH group. The mean HIT-6 score was 60.51 ± 5.84, the mean headache intensity was 7.15 ± 1.32, headache duration 11.00 ± 5.56 h, and headache frequency 11.71 ± 9.77 days a month. These results show that the participants in the TTH group had a very severe impact on their daily living tasks (Table 1).

### 3.3. Fat Intake

The fat intake was measured to see if there were differences between groups, and then, to establish correlations with the characteristics of the headache episodes in the TTH group. Neither the control group nor the TTH group had a recommended fat intake.

Comparing both groups, the TTH group had a significantly higher intake of saturated fat compared to the control group (34.40 ± 12.73 versus 20.74 ± 6.18 g; *p* < 0.01). This was the main difference between groups in fat intake.

There were no differences between groups regarding the intake of MUFAs (43.58 ± 17.65 versus 40.55 ± 13.99 g; *p* = 0.51), PUFAs (10.27 ± 3.21 versus 9.79 ± 6.42 g; *p* = 0.15), ω-6 fatty acids (6.86 ± 2.46 versus 6.56 ± 2.98 g; *p* = 0.44), ω-3 fatty acids (1.17 ± 0.39 versus 1.13 ± 0.75 g; *p* = 0.20), and ω-6: ω-3 ratio (6.33 ± 2.67 versus 7.08 ± 2.97; *p* = 0.33) (Table 1).

### 3.4. Correlations between the Continuous Variables

The correlations between the headache episodes and the fat intake of the TTHG were analyzed. Ω-6 fatty acid intake showed a positive moderate correlation with headache intensity (r = 0.48; *p* = 0.02). However, the ω-6: ω-3 ratio showed a negative moderate correlation with headache duration (r = −0.41; *p* = 0.04) and frequency (r = −0.42; *p* = 0.03). No other significant correlations were found.

## 4. Discussion

This novel research provides information about dietary habits in TTH patients, with special attention to the intake of fatty acids. The results were compared to the dietary habits of healthy patients without TTH participants who had a similar daily caloric intake. The TTH group reported significantly higher SFAs intake. In addition, significant associations were found between PUFAs intake and headache characteristics. As there were no significant differences between groups in age, BMI, and daily caloric intake, these mediating variables should not have influenced the results presented.

Participants in this study were middle-aged, slightly overweight, with a mean headache intensity of 7.15 ± 1.32, headache duration of 11.00 ± 5.56 h, and headache frequency of 11.71 ± 9.77 days per month, who reported a normal caloric intake of 2000 kcal per day. With respect to SFAs, the TTH group reported a significantly higher daily intake compared to the control group. In addition, the SFAs intake of the TTH was 34.40 g, equivalent to 310 kcal, or 14% of daily caloric intake, exceeding the health recommendations [29]. SFAs are mainly found in butter and red, processed meats [30]. According to Fritsche et al., excess SFAs intake could favor inflammatory reactions and could be responsible for causing headaches [31]. In a recent study, neuroinflammation was implicated in the pathogenesis of headache, as increased levels of proinflammatory cytokines, such as interleukin-1β and tumor necrosis factor-α are involved in the immune responses associated with headache [32]. These findings have been reported in several studies [17,33,34,35,36]. However, even though the TTH reported a significant impact of headaches on their daily living activities according to the HIT-6 questionnaire, there were no significant correlations between the HIT-6 score and fat intake.

Regarding the total daily intake of MUFAs, PUFAs, and ω-6 fatty acids, there were no significant differences between the TTH group and the control group. However, there was a positive moderate correlation between ω-6 fatty acid intake and headache intensity. In this line, dietary patterns with high consumption of ω-6 have been associated with an increased risk of suffering from diseases associated with inflammatory processes [14,31,37] and a higher prevalence of migraine headaches [10,16,17]. A possible explanation could be that the most common ω-6 fatty acid, linoleic acid, can give rise to arachidonic acid, which has inflammatory effects. Foods rich in ω-6 are oils derived from seeds (sunflower, corn, sesame), nuts (walnuts, pine nuts, peanuts, almonds, hazelnuts, and pistachios), quinoa, whole grains, meats (especially chicken and turkey), sausages, and eggs [14,31,38].

The TTH group and the control group showed similar intakes of ω-3 fatty acids, and this variable did not show any significant correlation with the headache episodes. However, the ω6: ω-3 ratio is more clinically relevant because if in the cell membranes the presence of ω-6 is well above the ω-3 fatty acids, there may be an inflammatory response. In fact, it is desirable that ω-6 and ω-3 fatty acids be consumed in similar amounts (ratio 1:1), although this is very rare. In the usual diet in the United States, this ratio is 20:1. Most experts recommend not exceeding a 5:1 ratio because, in that case, ω-6 fatty acids produce an inflammatory effect that can promote headaches, among other detrimental effects [10,39]. In the present study, in both groups, the ω-6: ω-3 ratio was higher than 5:1, without significant differences between groups. In a recent study, the incidence of headaches was lower in participants who increased the ω-3 fatty acid intake and simultaneously reduced the intake of ω-6 linoleic acid [17]. This effect could be explained because ω-6 fatty acids synthesize inflammatory hormones and because the excess of ω-6 prevents ω-3 fatty acids from fulfilling their functions [40]. Conversely, in the present study, there were moderate negative correlations between this fatty acid’s ratio and the duration (r = −0.41; *p* = 0.04) and frequency (r = −0.42; *p* = 0.03) of headache episodes.

Even though this is an observational study with a small sample size, the dietary advice is to increase ω-3 fatty acid intake, which are often lacking in the diet of the Western population, and to reduce ω-6 intake, which are often consumed in excess. Ω-3 fatty acids are abundant in oily fish, seafood, seeds, and nuts [30]. Regarding oily fish, sardines, mackerel, herring, or anchovies are more suitable than salmon, swordfish or tuna because larger species are more contaminated with mercury [41]. For these reasons, the Mediterranean dietary pattern would be advisable for these patients [40].

### 4.1. Clinical Implications and Future Lines of Research

This novel study provides useful information for health practitioners regarding the management of patients with TTH. An excess of SFAs intake could be a risk factor for developing chronic TTH, and an excess of ω-6 intake could have a negative impact on the intensity of headache episodes. Therefore, the diet should be assessed in patients with TTH.

Future randomized controlled trials should confirm these findings by carrying out dietary interventions in TTH patients, aiming to increase the intake of ω-3 fatty acids. Moreover, these studies should measure inflammatory markers and analyze possible associations with the intensity, duration, and frequency of headache episodes.

### 4.2. Limitations of the Study

The results of this study should be considered considering the following limitations: (i) The observational nature of this study does not allow for causality, (ii) all data were self-reported on dietary intake and may be subject to reporting bias. In addition, the study was carried out on 48 participants, which is a small sample size, so the results should be interpreted with caution.

## 5. Conclusions

Chronic tension-type headache patients showed higher intakes of SFAs when compared to controls. In addition, a higher ω-6 fatty acid intake was associated with more severe headache episodes. Therefore, the characteristics of the diet, in particular the dietary fatty acid intake, should be considered in the treatment of patients with TTH. A healthy diet could lead to less severe headache episodes. Future research should confirm these findings by carrying out dietary interventions in TTH patients.

## Figures and Tables

**Figure 1 jcm-11-07139-f001:**
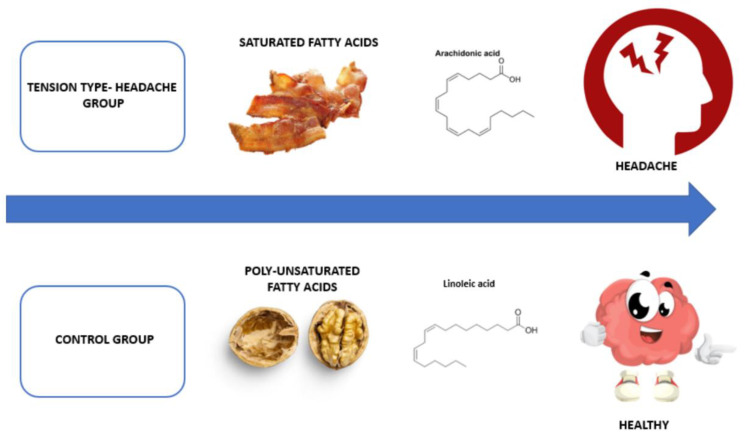
Relationship between saturated and poly-unsaturated fatty acids with tension-type headache.

**Table 1 jcm-11-07139-t001:** Comparative analysis of daily intake of fats between tension-type headache group (*n* = 24) and control group (*n* = 24).

Variables	Tension-Type Headache Group Mean ± SD	Control Group Mean ± SD	*p*-Value
**Age (years)**	37.58 ± 13.36	42.57 ± 8.11	0.12
**Body Mass Index (kg/m^2^)**	25.18 ± 4.62	24.99 ± 3.95	0.88
**Kcal**	2175.08 ± 519.58	1960.48 ± 403.16	0.11
**SFAs (g)**	34.40 ± 12.73	20.74 ± 6.18	<0.01 *
**MUFAs (g)**	43.58 ± 17.65	40.55 ± 13.99	0.51
**PUFAs (g)**	10.27 ± 3.21	9.79 ± 6.42	0.15
**ω-6 fatty acids (g)**	6.86 ± 2.46	6.56 ± 2.98	0.44
**ω-3 fatty acids (g)**	1.17 ± 0.39	1.13 ± 0.75	0.20
**ω-6: ω-3 ratio**	6.33 ± 2.67	7.08 ± 2.97	0.33

Significance * was set at an alpha level of <0.05.

## Data Availability

The data are available upon request from the corresponding author.

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
