# Peer review of "Associations between Fatty Acid Intake and Tension-Type Headache: A Cross-Sectional Study"

_jcm, 2022, doi:10.3390/jcm11237139_

Round 1

Reviewer 1 Report

Comments:

The study is a cross sectional study evaluating association between fatty acid intake and tension type headache. The research question is clinically important, however, there are several points which needs to be addressed.

1-    The major problem with the manuscript is the sample size. How the authors arrived at the figure or 24 in each arm? It would be erroneous to conclude any scientific association or hypothesis with such a small sample size of 24 especially when the authors have mentioned the prevalence of TTH to be 36-78%.

2-    The diagnosis of Chronic TTH was made as per criteria of Headache study group of Spanish Society of Neurology or HIS classification? Under the methods section, both have been mentioned. This should be clarified.

3-    Authors should mention the study period.

4-    No mention of exclusion criteria. Were the patients with co-existing other neurological diseases, like stroke, epilepsy, brain tumors, head injury etc included?

5-    Use of Food Frequency questionnaires have their short comings. It could have led to recall bias. Did the authors looked at the amount of fat, duration of the diet, other component of diet like Carbohydrate and protein?

6-    Were the baseline parameters matched for co morbidities like DM, HTN, smoking, alcohol etc ? No mention of these parameters in the manuscript.

7-    Were the factors like anxiety, depression, sleep abnormalities and other lifestyle factors which are very common among patients with CTTH matched at baseline? No mention of any such analysis is mentioned in the manuscript.

8-    The authors have not discussed about any confounder in the study. Which I think is another major drawback of the study.

9-    Clinical information like, disease duration has not been mentioned.

10- In the results section, while the authors focused on the saturated fatty acids, there is no data about Total Fat intake in these two groups. It would have been interesting if the authors had looked at other nutrients also.

11- Table 1 doesn’t contain any information about headache characteristics or HIT-6.

Author Response

Author's response: Thank you very much for reviewing our manuscript and for the evaluations made, without a doubt the comments give us a lot of value, especially to continue improving in our research career, we will immediately make all the modifications to be able to carry out an improvement of the work, thank you very much from of the entire research team.

Specifically, we respond to each of the comments:

Reviewer #1:

The study is a cross sectional study evaluating association between fatty acid intake and tension type headache. The research question is clinically important, however, there are several points which needs to be addressed.

Methodology

  • The major problem with the manuscript is the sample size. How the authors arrived at the figure or 24 in each arm? It would be erroneous to conclude any scientific association or hypothesis with such a small sample size of 24 especially when the authors have mentioned the prevalence of TTH to be 36-78%.

Author's response: Thank you for pointing this out. The reviewer is correct, and we have corrected this issue by adding the sample size calculation and the following reference for prevalence data (Stovner, L.J., Hagen, K., Linde, M. et al. The global prevalence of headache: an update, with analysis of the influences of methodological factors on prevalence estimates. J Headache Pain 23, 34 (2022). https://doi.org/10.1186/s10194-022-01402-2)

  • The diagnosis of Chronic TTH was made as per criteria of Headache study group of Spanish Society of Neurology or HIS classification? Under the methods section, both have been mentioned. This should be clarified.

Author's response: As suggested by the reviewer, we have clarified these criteria.

  • Authors should mention the study period.

Author's response: Thank you for pointing this out. We have mentioned the study period

  • No mention of exclusion criteria. Were the patients with co-existing other neurological diseases, like stroke, epilepsy, brain tumors, head injury etc included?

Author's response: Thank you for this suggestion. We agree with the reviewer’s assessment and eligibility criteria section was improved.

  • Use of Food Frequency questionnaires have their short comings. It could have led to recall bias. Did the authors looked at the amount of fat, duration of the diet, other component of diet like Carbohydrate and protein?

Author's response: Thank you for pointing this out. We have analyzed our data according to the described (see reference) relationship between fatty acids and headache (Razeghi Jahromi S, Ghorbani Z, Martelletti P, Lampl C, Togha M; School of Advanced Studies of the European Headache Federation (EHF-SAS). Association of diet and headache. J Headache Pain. 2019 Nov 14;20(1):106. doi: 10.1186/s10194-019-1057-1. PMID: 31726975; PMCID: PMC6854770.)

  • Were the baseline parameters matched for co morbidities like DM, HTN, smoking, alcohol etc ? No mention of these parameters in the manuscript.

Author's response: Thank you for pointing this out. The reviewer is correct, and we have corrected this issue in participants’ section.

  • Were the factors like anxiety, depression, sleep abnormalities and other lifestyle factors which are very common among patients with CTTH matched at baseline? No mention of any such analysis is mentioned in the manuscript.

Author's response: Thank you for pointing this out. The reviewer is correct, and we have corrected this issue in participants’ section.

Discussion

  • The authors have not discussed about any confounder in the study. Which I think is another major drawback of the study.

  • Clinical information like, disease duration has not been mentioned.

Author's response: Thank you very much for the comment, within the methodology we have added this detail in the Participant part:

Participants

Recruitment of participants was carried out (between September and December 2021) among students and workers at the European University of Madrid. 48 participants were recruited for the study, 24 were selected for the control group and 24 were selected to be part of the chronic TTH group, this last group, previously attended their medical center, where they were diagnosed with TTH by their neurologist, following the headache classification criteria of the HIS classification, in its third edition (20). The diagnostic criteria followed were: (i) headache occurs on average more than 15 days per month for more than 3 months (more than 180 days per year), (ii) with a duration of minutes to days or without remission…

In addition, we have included a paragraph within the Discussion to clarify the comment in detail:

Participants in this study were middle-aged, slightly overweight, with a mean headache intensity of 7.15 ± 1.32, headache duration of 11.00 ± 5.56 hours, and headache frequency of 11.71 ± 9.77 days per month, who reported a normal caloric intake of 2000 kcal per day.

Reviewer 2 Report

Domínguez-Balmaseda et al. compare dietary fatty acid intake between participants with and without chronic TTH and investigate the cross-sectional association of dietary fatty acid intake, pain characteristics, and quality of life.

It was very interesting reading, but I believe there are a number of issues that should be improved.

1) To better understand readers, please define the abbreviation when it first appears. Verify this information in all the main text. Abstract. STROBE, CT, SFA, PUFA should be described.

2) Line 31. Please, provide the references in main document according to the authors instructions.

3) Line 47. Please homogenize the abbreviature. In the abstract, it was shown as SFA.

4) Please describe Figure 1 in the main document. Appropriate information should be provided.

5)Figure 2 should be provided indicating the methodology. In these forms as showed is confusing for the reader.

6) Figure 1 legend. More information about the figure should be provided.

7) Lines 87-90. Please clarify this sentence. It is difficult to understand. Besides, the control group and the number of participants in each group needs to be described.

Line 95. please verify m2. Is it correct?

8) Methodology is missing to describe the control group. Also, how many participants are in the control group and TTHG. The inclusion and exclusion criteria must be provided. Furthermore, the authors were careful to exclude patients with certain diseases.

Finally, a description of the frequency of application of the questionnaires (CFCA) should be provided, and how they classified the high intake of SFAs (g), MUFAs (g), PUFAs (g), ω-6 fatty acids (g), and ω-3 fatty acids (g).

9) Results section.

It is necessary that a short introduction and conclusion in each section will be provided before describing all the experiments and results that support each section. It is to guide the reader to understand the importance of the manuscript

10) Headache’s episodes: Headache’s episodes are not contained in Table 1. Besides, the Headache Impact Test (HIT-6) measures was applied to the control group? if so, these data must be provided

11) Fat intake: Authors should discuss whether participants in the control group had a recommended intake

12) Table 1. The authors must explain why the control group present a higher value in ω-6: ω-3 ratio. Since it has been previously reported that an increase in this radius leads to   Tension-type headache.

13) Table 1. indicates what’s mean *. Besides, the intake reported in table 1 to what frequency does it correspond? (Daily, weekly, monthly).

14) Discussion section.

It is suggested that the authors present the writing as results and discussion. The information provided in the discussion section will help support the results obtained and provide a better understanding of the manuscript. In this format, as it is presented, the document is confusing, vague, and contradictory.

15) Limitations of the study

Is necessary to describe if the participant´s numbers are correct (n=24) and if it is applies  to  all populations.

16) Conclusions do not reflect the results of the study because the authors conclude that "Ω-227 6 fatty acids intake and ω-6: ω-3 ratio were associated with headache episodes" (lines 227-228). However, in the control group the ω-6: ω-3 ratio was higher compared to the Tension-type headache group.

Author Response

Author's response: Thank you very much for reviewing our manuscript and for the evaluations made, without a doubt the comments give us a lot of value, especially to continue improving in our research career, we will immediately make all the modifications to be able to carry out an improvement of the work, thank you very much from of the entire research team.

Specifically, we respond to each of the comments:

Comments RR2:

Domínguez-Balmaseda et al. compare dietary fatty acid intake between participants with and without chronic TTH and investigate the cross-sectional association of dietary fatty acid intake, pain characteristics, and quality of life.

It was very interesting reading, but I believe there are a number of issues that should be improved.

Introduction

1) To better understand readers, please define the abbreviation when it first appears. Verify this information in all the main text. Abstract. STROBE, CT, SFA, PUFA should be described.

Author response: Thanks for the comment, we have improved and added the meaning of the mentioned acronyms.

 2) Line 31. Please, provide the references in main document according to the authors instructions.

Author response: Thanks for the comment, we have checked all the references

3) Line 47. Please homogenize the abbreviature. In the abstract, it was shown as SFA.

Author response: Thanks for the comment, we have reviewed all the acronyms for their correct writing

4) Please describe Figure 1 in the main document. Appropriate information should be provided.

Author response: Thanks for the comment, we have implemented the description of the figure as well as in the text, referring to the drawing.

Methodology

5) Figure 2 should be provided indicating the methodology. In these forms as showed is confusing for the reader.

Author response: Thank you for pointing this out. We have removed Figure 2.

6) Figure 1 legend. More information about the figure should be provided.

Author's response: Thank you for comment. We have improved the figure 1 legend.

7) Lines 87-90. Please clarify this sentence. It is difficult to understand. Besides, the control group and the number of participants in each group needs to be described.

Author's response: Thank you for comment. We have improved this sentence.

Line 95. please verify m2. Is it correct?

Author's response: Thank you for comment. BMI = Kg/m2. Quetelet’s Body Mass Index

8) Methodology is missing to describe the control group. Also, how many participants are in the control group and TTHG. The inclusion and exclusion criteria must be provided. Furthermore, the authors were careful to exclude patients with certain diseases.

Author's response: Thank you for pointing this out. The reviewer is correct, and we have corrected this issue by adding the sample size calculation and improving the eligibility criteria.

Finally, a description of the frequency of application of the questionnaires (CFCA) should be provided, and how they classified the high intake of SFAs (g), MUFAs (g), PUFAs (g), ω-6 fatty acids (g), and ω-3 fatty acids (g).

Author's response: Thank you for comment. We have improved this section.

Results section.

It is necessary that a short introduction and conclusion in each section will be provided before describing all the experiments and results that support each section. It is to guide the reader to understand the importance of the manuscript

Author’s response: We really appreciate your comment. We have included a short introduction and conclusion, to provide a better understanding of the manuscript.

10) Headache’s episodes: Headache’s episodes are not contained in Table 1. Besides, the Headache Impact Test (HIT-6) measures was applied to the control group? if so, these data must be provided

Author’s response: The HIT-6 was not applied to the control group, as it is a scale for headaches, and the control group were healthy participants. The headache’s episodes are not contained in Table 1, because it only implies the TTH group, and Table 1 is focused on nutrients daily intake.

11) Fat intake: Authors should discuss whether participants in the control group had a recommended intake

Author’s response: Thank you. We have clarified it.

12) Table 1. The authors must explain why the control group present a higher value in ω-6: ω-3 ratio. Since it has been previously reported that an increase in this radius leads to   Tension-type headache.

Author’s response: Thank you for your question. Even if the control group presented a higher value in this variable, this difference was not statistically significant. So we do not consider it relevant. Besides, both groups should unhealthy values in the ω-6: ω-3 ratio.

13) Table 1. indicates what’s mean *. Besides, the intake reported in table 1 to what frequency does it correspond? (Daily, weekly, monthly).

Author’s response: We appreciate your comment. As indicated in the title of Table 1, it corresponds to daily intake. We have clarified what * means.

Discussion section.

It is suggested that the authors present the writing as results and discussion. The information provided in the discussion section will help support the results obtained and provide a better understanding of the manuscript. In this format, as it is presented, the document is confusing, vague, and contradictory.

Author's response: Thank you very much for the comment. We have tried to improve both sections in the discussion, hoping to improve the manuscript.

15) Limitations of the study

Is necessary to describe if the participant´s numbers are correct (n=24) and if it is applies  to  all populations.

Author's response: Thank you very much for the comment. The participant’s number was 48. We have clarified it and commented about it as a limitation.

16) Conclusions do not reflect the results of the study because the authors conclude that "Ω-227 6 fatty acids intake and ω-6: ω-3 ratio were associated with headache episodes" (lines 227-228). However, in the control group the ω-6: ω-3 ratio was higher compared to the Tension-type headache group.

Author’s response: Thank you very much for your comment. In the present study, in both groups the ω6: ω-3 ratio was higher than 5:1, without statistically significant differences between groups.

The TTGH had an unhealthy ω-6: ω-3 ratio, which in addition showed negative correlations with the severity of the headaches.

Reviewer 3 Report

The aims of this  observational study were to compare dietary fatty acids intake between participants with and without chronic TTH and to investigate the association between dietary fatty  acid intake, pain characteristics, and quality of life in patients with chronic TTH.

The research topic is quite original and very interesting for its clinical implications. The language of the paper  is understandable. The methods are well described.

However, the sample of patients is rather small, as stated by the Authors themselves among the limitations of the study.

Authors should kindly specify whether patients and controls were sex matched and whether patients had TTH associated or not associated with pericranial tenderness. 

Another aspect that should be clarified is why in the study sample "headache intensity was 7.15 ± 1.32"  if  the pain of TTH is by definition mild or moderate.

Finally, the Authors should explain if patients with TTH were taking nutraceuticals,  symptomatic and / or preventive drugs and if they were treated with non-drug therapy. this specification should be made because some drugs and nutraceuticals affect lipid metabolism and body mass index.

Results are sound   are discussed exhaustively. 

In summary Chronic tension-type headache patients showed higher intakes of SFAs when compared to controls. In addition, a higher ω -6 fatty acids intake was associated with more severe headache episodes.

Despite the limitations and the low number of patients, this study appears in my opinion worthy of publication, because it contains an important message for the clinician and that is to  analyze the type of nutrition of the patients in search of all possible factors that can affect the course of headache.

Author Response

The aims of this  observational study were to compare dietary fatty acids intake between participants with and without chronic TTH and to investigate the association between dietary fatty  acid intake, pain characteristics, and quality of life in patients with chronic TTH.

The research topic is quite original and very interesting for its clinical implications. The language of the paper  is understandable. The methods are well described.

Thank you very much for these comments.

However, the sample of patients is rather small, as stated by the Authors themselves among the limitations of the study.

Totally agree, in the future we would like to carry out the study with a larger sample size and with different centers

Authors should kindly specify whether patients and controls were sex matched and whether patients had TTH associated or not associated with pericranial tenderness.

We leave this text in the manuscript to contextualize this answer:

The diagnostic criteria followed were: (i) headache occurs on average more than 15 days per month for more than 3 months (more than 180 days per year) with pericranial tenderness in neck and face region,

Individuals with TTH were matched with their healthy controls by gender, age, body mass index (BMI) and nutritional energy intake (kcal).

Another aspect that should be clarified is why in the study sample "headache intensity was 7.15 ± 1.32"  if  the pain of TTH is by definition mild or moderate.

We have added this sentence to clarify:

These results show that the participants in the TTH group had very severe impact on their daily living tasks

Finally, the Authors should explain if patients with TTH were taking nutraceuticals, symptomatic and / or preventive drugs and if they were treated with non-drug therapy. this specification should be made because some drugs and nutraceuticals affect lipid metabolism and body mass index.

Thank you very much for the comment. We have added in the exclusion criteria that the study participants did not take nutraceuticals:

The exclusion criteria for both groups were: neurological, metabolic diseases such as diabetes mellitus, cardiovascular diseases such as arterial hypertension or severe respiratory diseases, whiplash injury, neck fracture, mental health disease (anxiety or depression) assessed with Hospital Anxiety and Depression Scale (>8 points in each subscale) (21) and taking nutraceuticals.

Results are sound   are discussed exhaustively.

In summary Chronic tension-type headache patients showed higher intakes of SFAs when compared to controls. In addition, a higher ω -6 fatty acids intake was associated with more severe headache episodes.

Despite the limitations and the low number of patients, this study appears in my opinion worthy of publication, because it contains an important message for the clinician and that is to  analyze the type of nutrition of the patients in search of all possible factors that can affect the course of headache.

Round 2

Reviewer 1 Report

Comments:

The authors have not addressed my queries adequately. The main issue is very small sample size. Only 24 cases. This can at best qualify for a pilot study. Here is my detailed replay:

1-      Sample size: In the version1, the authors mentioned the prevalence of TTH to be between 36-78%. However, the authors have changed the figure in the Version 2 to 15.8%. Strangely, the reference provided by the authors report the prevalence of TTH to be 26 % (Stovner et al, available online from April 2022). This amounts to misrepresentation of data. Moreover, how the sample size calculation was based on this study (as mentioned in the reply), when the Stovner et al paper was published in April 2022, and the current study was conducted between September to December 2021. This indicates that the sample size calculation was not done prior to the study and it seems it was in response to the queries raised in 1st review. One component of STROBE check list is study size. However, here it seems the authors have not adhered to STROBE in spirit. This is not acceptable.

2-      In the sample size calculation section, the authors have mentioned about a Pilot study, but the reference provided (Ref 26) is of G-power statistics. The authors have not mentioned any data from the pilot study. Moreover, prevalence of TTH has not been taken in to account. This seems more like a post- review exercise to get to the number of 48!

3-      Page 2, line 59-61. The newly added statement is completely wrong interpretation of the quoted reference.

4-      How does the aim of the study changes between two submissions? In first, the authors looked at Pain characteristics, and quality of life. But in second they mention severity of headaches.

5-      Baseline parameters like DM, HTN have not been mentioned in the patients section, as has been replied by the authors. Authors have stated that anxiety, depression were excluded, but how they were excluded (which scales were used) has not been mentioned. The prevalence of anxiety is about 20% in chronic TTH (Song et al 2016), so it is very important to state how they were excluded.

6-      Authors should refrain from making any dietary advices (page 7, line 250-256) based  on such a small study

In the results and discussion, the authors state that omega- 6: Omega 3 ratio had a negative correlation with headache frequency and duration. This means an increased ratio (or an increased omega 6) will reduce the frequency and headache duration ?

Author Response

  • Sample size: In the version1, the authors mentioned the prevalence of TTH to be between 36-78%. However, the authors have changed the figure in the Version 2 to 15.8%. Strangely, the reference provided by the authors report the prevalence of TTH to be 26 % (Stovner et al, available online from April 2022). This amounts to misrepresentation of data. Moreover, how the sample size calculation was based on this study (as mentioned in the reply), when the Stovner et al paper was published in April 2022, and the current study was conducted between September to December 2021. This indicates that the sample size calculation was not done prior to the study and it seems it was in response to the queries raised in 1st One component of STROBE check list is study size. However, here it seems the authors have not adhered to STROBE in spirit. This is not acceptable.

Author's response:  We would like to apologize below for having made a mistake in the data, we have revised the data thoroughly so as not to confuse the reviewers and we have modified the data in the manuscript, we add the paragraph to:

Tension-type headache (TTH) is characterized by recurrent headaches that can become disabling, often accompanied by sensory, motor and neurovascular alterations, with a prevalence of 26% in general population. (1,2).

The sample size calculation was performed with data from our pilot study (n=10). To the authors’ knowledge, this method has been performed and validated in several studies according to STROBE check list (See references below).

Almazán-Polo J, López-López D, Romero-Morales C, Rodríguez-Sanz D, Becerro-de-Bengoa-Vallejo R, Losa-Iglesias ME, Bravo-Aguilar M, Calvo-Lobo C. Quantitative Ultrasound Imaging Differences in Multifidus and Thoracolumbar Fasciae between Athletes with and without Chronic Lumbopelvic Pain: A Case-Control Study. J Clin Med. 2020 Aug 14;9(8):2647. doi: 10.3390/jcm9082647. PMID: 32823967; PMCID: PMC7464501.

Calvo-Lobo C, Useros-Olmo AI, Almazán-Polo J, Becerro-de-Bengoa-Vallejo R, Losa-Iglesias ME, Palomo-López P, Rodríguez-Sanz D, López-López D. Rehabilitative ultrasound imaging of the bilateral intrinsic plantar muscles and fascia in post-stroke survivors with hemiparesis: A case-control study. Int J Med Sci. 2018 Jun 4;15(9):907-914. doi: 10.7150/ijms.25836. PMID: 30008603; PMCID: PMC6036101.

  • In the sample size calculation section, the authors have mentioned about a Pilot study, but the reference provided (Ref 26) is of G-power statistics. The authors have not mentioned any data from the pilot study. Moreover, prevalence of TTH has not been taken in to account. This seems more like a post- review exercise to get to the number of 48!

Author's response:  We are sorry about the misleading. We have added the mean and standard deviation of each group.

  • Page 2, line 59-61. The newly added statement is completely wrong interpretation of the quoted reference.

Author's response:  Thank you very much for the appreciation, we had an error with the reference, now it is correct. In addition, we have added a small modification to the text that we highlight below, highlighting that the intake of polyunsaturated fatty acids could improve the prevalence of tension headaches:

on the contrary, an adequate intake of polyunsaturated fatty acids could lead to an improvement in the quality of life of patients with TTH, preventing the headache or reducing it considerably (9), as well as the intensity (18).

  • How does the aim of the study changes between two submissions? In first, the authors looked at Pain characteristics, and quality of life. But in second they mention severity of headaches.

Author's response:  We are sorry about the misleading. We did not mean to change the aim of the study. but to write it better so I could be more understandable. According to your comment, we have reformulated the objective of the study as it was in the original version.

  • Baseline parameters like DM, HTN have not been mentioned in the patients section, as has been replied by the authors. Authors have stated that anxiety, depression were excluded, but how they were excluded (which scales were used) has not been mentioned. The prevalence of anxiety is about 20% in chronic TTH (Song et al 2016), so it is very important to state how they were excluded.

Author's response:  The exclusion criteria for both groups were: neurological, metabolic diseases such as diabetes mellitus, cardiovascular diseases such as arterial hypertension or severe respiratory diseases, whiplash injury, neck fracture, mental health disease (anxiety or depression).

Anxiety and depression were assessed with the Hospital Anxiety and Depression Scale (see reference 21. Zigmond, 1983).

6-      Authors should refrain from making any dietary advices (page 7, line 250-256) based  on such a small study

In the results and discussion, the authors state that omega- 6: Omega 3 ratio had a negative correlation with headache frequency and duration. This means an increased ratio (or an increased omega 6) will reduce the frequency and headache duration ?

Author's response:  Thank you very much for your comment on the negative correlation between the omega- 6: Omega 3 ratio and the headaches’ frequency and duration. We have now corrected the interpretation.

Concerning the dietary advice, we agree that it is a small study, but inflammation has a certain role in the pathogenesis of TTH, and the fatty acids intake influences inflammation. We are advising a healthy diet pattern, which has proved to provide anti-inflammatory effects.

Reviewer 2 Report

The manuscript is substantially improved

Author Response

(The authors gave the same response as above.)
